# Epidemiology of *Nosema* spp. and the effect of indoor and outdoor wintering on honey bee colony population and survival in the Canadian Prairies

**Rosanna N. Punko** **[1]\*, Robert W. Currie[1], Medhat E. Nasr[2], Shelley E. Hoover**[3]

**1** Department of Entomology, University of Manitoba, Winnipeg, Manitoba, Canada, **2** Alberta Agriculture and Forestry, Government of Alberta, Edmonton, Alberta, Canada, **3** Department of Biological Sciences, University of Lethbridge, Lethbridge, Alberta, Canada

\* punkor@myumanitoba.ca

## Abstract

The epidemiology of *Nosema* spp. in honey bees, *Apis mellifera*, may be affected by winter conditions as cold temperatures and differing wintering methods (indoor and outdoor) provide varying levels of temperature stress and defecation flight opportunities. Across the Canadian Prairies, including Alberta, the length and severity of winter vary among geographic locations. This study investigates the seasonal pattern of *Nosema* abundance in two Alberta locations using indoor and outdoor wintering methods and its impact on bee population, survival, and commercial viability. This study found that *N. ceranae* had a distinct seasonal pattern in Alberta, with high spore abundance in spring, declining to low levels in the summer and fall. The results showed that fall *Nosema* monitoring might not be the best indicator of treatment needs or future colony health outcomes. There was no clear pattern for differences in *N. ceranae* abundance by location or wintering method. However, wintering method affected survival with colonies wintered indoors having lower mortality and more rapid spring population build-up than outdoor-wintered colonies. The results suggest that the existing *Nosema* threshold should be reinvestigated with wintering method in mind to provide more favorable outcomes for beekeepers. Average *Nosema* abundance in the spring was a significant predictor of end-of-study winter colony mortality, highlighting the importance of spring *Nosema* monitoring and treatments.

## Introduction

*Nosema apis* Zander and *Nosema ceranae* Fries et al. are spore-forming obligate parasites of the midgut epithelial cells of honey bees. Initially, *Nosema* infection in *Apis mellifera* Linnaeus was only caused by *N. apis*, but in 2006, *N. ceranae* was also identified in *A. mellifera* [1], and now both species are prevalent in honey bees worldwide [2]. *Nosema ceranae* has become the predominant species in many regions, suggesting that it is replacing *N. apis* [2–5]. In Europe, the proportion of *N. ceranae* infections appears to be greater in warmer climates than

**Data Availability Statement:** All relevant data are within the Supporting Information files.

**Funding:** Alberta Crop Industry Development Fund (ACIDF) M.N. Alberta Beekeepers Commission R.C.

https://www.albertabeekeepers.ca/ Growing Forward 2 (a federal-provincial-territorial initiative) M.N University of Manitoba R.C. https://umanitoba.ca/ Canadian Bee Research Fund (CBRF) R.C., S.H. https://honeycouncil.ca/industry-overview/canadian-bee-research-fund/ The funders had no role in study design, data collection and analysis, decision to publish, or preparation of the manuscript.

**Competing interests:** The authors have declared that no competing interests exist.

temperate climates [6]. Regional differences in the relative dominance of *N. ceranae* may be due to *N. ceranae* tolerating higher temperatures than *N. apis*, whereas *N. apis* is more cold-tolerant [6–9]. However, *N. ceranae* is also successful in cold climates, having become the most prevalent species in Canada and Siberia, suggesting temperature is not as important as previously thought [10–16].

*Nosema* infection has been shown to affect colony strength and productivity, with *N. apis* associated with reduced bee populations and brood and honey production as well as increased winter losses [17–20]. However, the effect of *N. ceranae* on the colony is less clear. Higher proportions of infected foragers have been negatively correlated with brood production and worker bee population in Spain [21,22]. Multiple studies have found *N. ceranae* to be associated with sudden depopulation and colony death [21–23]. In contrast, a long-term study in Germany found that *N. ceranae* infections in the spring or fall were not correlated with colony losses in the following summer or winter [24]. A similar lack of effects on colony strength or winter mortality was reported in Manitoba [25] and Nova Scotia, Canada [26]. However, a recent study from Ontario found that *N. ceranae* was negatively correlated with bee population and food stores, but not colony mortality [27]. More research is needed to understand the effect of *N. ceranae* on honey bee colonies under various regional climatic and seasonal conditions and the variability observed in colony-level effects.

Awareness of the seasonal trends of parasites informs the appropriate timing of treatments to prevent outbreaks. *Nosema apis* has a well-established seasonal pattern in honey bees with the highest spore abundance in the spring, lower spore abundance in summer, and typically another smaller peak in the fall followed by a slow increase over the winter in colonies wintered outdoors [11,28,29]. In contrast, studies to date on *N. ceranae* have shown high variability and seasonality that is difficult to predict [11,21,27,30,31]. Data on *Nosema* seasonality in Canada's prairie region are needed to inform the management of this pathogen.

*Nosema* can be controlled using fumagillin, a product registered in Canada under the trade name, Fumagilin-B®. Alberta's current control recommendations are to apply fumagillin in the spring and fall when spore abundance is above one million spores per bee [32]. However, this nominal threshold used throughout North America was established for *N. apis* infections and has not been appropriately validated for either *N. apis* or *N. ceranae* under different beekeeping winter management and climatic conditions.

Due to its vast area and geographical features, Alberta has a wide range of bioregions across the province that differ in climate and vegetation. The majority of managed colonies in Alberta are located in the Parkland and Grassland bioregions [33]. The Grassland bioregion in southern Alberta is the hottest and driest region and experiences milder winters due in part to the winter Chinook (warming) winds that extend as far north as Red Deer, AB [34]. The Parkland bioregion in central Alberta has a shorter growing season and a lower mean annual temperature by 1.7°C than the Grassland [34]. The warmer temperatures in the Grassland may reduce *Nosema* abundance due to decreased winter temperature stress on the honey bees. Additionally, both the Grassland and Parkland are heavily cultivated, with natural vegetation dominated by grasses in the Grassland and aspen trees in the Parkland zone [34]. The lack of tree windbreaks in the Grassland may affect outdoor wintering survival, and beekeepers often add a windbreak if shelter is not available.

Honey bee diseases can become particularly detrimental to colonies in winter [12]. Bees typically defecate while flying when the ambient temperature is above 10°C [35]. Winter temperatures prevent bees from defecating through such "cleansing" flights, causing bees to hold their feces within their rectum for long periods of cold weather, increasing *Nosema* spore abundance in the gut [35]. Eventually, bees may be forced to defecate within the hive, which could increase the spread of infection [35]. In Canada, colonies can be either wintered

outdoors or indoors. Colonies that overwinter outdoors are wrapped with insulating covers to protect them from the elements and trap heat, but bees can still enter and exit the hive on warmer days. In contrast, colonies that overwinter indoors are moved in the autumn (late October) into buildings that are temperature regulated at 4–5˚C with constant air exchange and air remixing where the colonies are always kept in the dark to prevent bee flight [36]. Therefore, different wintering management options provide varied defecation potential and exposure to a range of temperatures during the long Canadian Prairie winters.

How these differences in climate and management may affect *Nosema* is poorly understood. Outdoor wintering may allow for defecation flights, thus reducing their *Nosema* spore load. Additionally, the intermittent short periods of warm weather brought to southern Alberta by Chinook winds could provide more opportunities for winter defecation events than in central Alberta, which typically has consistently cold temperatures throughout the winter. However, indoor wintering may reduce stress, a factor that often exacerbates disease, by avoiding extreme temperature fluctuations.

The objectives of this study were to: (1) characterize the patterns of seasonal variation in *Nosema ceranae* abundance in honey bee colonies over two years; (2) assess the pattern of *Nosema* abundance in different climatic zones within Alberta (Parkland and Grassland biore-gions); (3) assess the impact of different wintering management methods (indoors versus out-doors) on *Nosema*, and (4) characterize the impact of variation in *Nosema* in these environments on honey bee colony population, survival, and commercial viability.

## Methods

### Experimental design

The study ran continuously from June 2017 to April 2019 and spanned two winters. Honey bee colonies borrowed from local beekeepers were located in two apiaries near Edmonton, Alberta (53˚38'49.5 "N 113˚21'25.5 "W and 53˚39'32.2 "N 112˚38'38.2 "W), and in two apiaries near Rainier, Alberta (50˚22'33.6 "N 112˚05'19.3"W and 50˚23'50.0"N 112˚06'41.6"W), which are within the Parkland and Grassland bioregions of Alberta, respectively [34]. Hereafter, the apiaries near Edmonton will be referred to as "North" (as it is North of Rainier) and the Rain-ier apiaries as "South". These locations were separated by 370 km. Within regions, North api-aries were 47 km apart, and South apiaries were 3 km apart. In each apiary, of approximately 40 honey bee colonies, eight colonies were randomly selected for this study.

At the beginning of the study, the adult bee and brood population were equalized for all 40 colonies in each apiary to make them as similar in size as possible. This equalization occurred from May 29–30 in the North apiaries and June 7–8 in the South apiaries. Several days before equalization, existing queens were removed from the colonies. On the day of equalization, adult bees from all colonies were shaken into a large, screened box (52" L x 24" W x 28.5" H). Next, all available brood, pollen, and honey frames were shared equally among the colonies. Then, a scoop was used to distribute the bees equally among the colonies. The colonies were fed 3.8 L of 2:1 sugar syrup from in-hive feeders, and the entrances were screened for 2–3 days to ensure the bees were retained in their new hives. All colonies were given new mated queens (Kona Queen Hawaii, USA) 1–3 days post-equalization, all of which were marked with a paint dot on their thorax. After equalization, each apiary had 40 equivalent single chamber colonies with newly mated queens. Additional brood chambers containing empty frames and honey and pollen frames were added when the bee population became too large for a single brood chamber.

In each apiary, eight colonies were selected for this longitudinal study, for a total of 32 colo-nies. One colony was removed from the study entirely as the colony never accepted the

introduced queen, and the colony population collapsed before the second sampling date. Therefore, the study started with 31 colonies. Colonies were not treated with fumagillin for the entirety of the study. For winter 2017–2018, one apiary was wintered outdoors, and the other apiary was wintered indoors at each of the two locations. In mid to late October, the outdoor-wintered colonies were wrapped with a commercial western insulated wrap with a top pillow, usually in groups of four on a pallet as per standard practice for the region [37]. When only three colonies were on a pallet, a stack of empty boxes was used to stand in for the fourth colony so that the wrap fits properly. Colonies were provided with a top entrance hole in the front for ventilation, which allowed the bees to exit and re-enter the hive. Pallets of colonies that were wintered indoors were transported approximately 13 km from the apiary site to a wintering building in each location and stored in stacks of five high in several rows, along with non-experimental colonies from the beekeeping operation. The wintering buildings were maintained at 4–5˚C with a ventilation rate of 0.25 L/s per colony (in the wintering building) and were kept dark to prevent bee flight [36]. For winter 2018–2019, each remaining live colony received the same wintering treatment that it had previously. Additionally, if a colony became queenless during the study, this was noted and it was given a mated replacement queen (Kona Queen Hawaii, USA).

Due to the distance between locations, the North and South were sampled/evaluated on alternating weeks with sampling periods ranked in ascending order within locations (e.g. first sample taken = 1, second sample taken = 2, etc.). Sampling periods in different locations with the same rank were considered sampled at the same time. For example, North sample 1 on June 12–13, 2017 and South sample 1 on June 15–16, 2017 were grouped. Within a location, apiaries were sampled on the same day or one day later. For ease of reference and analysis, dates are presented as the average date of the sampling periods within the same rank (e.g. sample 1 for both locations is dated June 14, 2017). Table A (in S1 File) shows the timeline for colony sampling and bee population evaluations.

*Varroa* mite populations were monitored throughout the study and maintained below the economic threshold (3%). *Varroa* was controlled in all colonies with Apivar® (500 mg Amitraz/strip) at the beginning of September in 2017 and 2018 as some colonies had infestation levels above the 3 mites per 100 bees fall threshold [38].

### *Nosema* abundance

Worker bees were also collected approximately every two weeks for *Nosema* analysis, beginning in June 2017 until colonies were wintered and the same sampling regime resuming following the winter. For these samples, approximately 100 adult bees were collected from either the outer honey frames in the brood chamber or from honey supers [39]. For the first sample following winter, colonies that died over the winter were also sampled for *Nosema* but by collecting dead bees from the bottom boards. Bee samples were stored in 70% ethanol at room temperature until further processing. Samples were prepared for analysis by grinding the abdomens of 30 bees with 5 mL water in a 35 mL conical tissue grinder. The solution was then poured into a new 50 mL conical tube. An additional 10 mL of water was used to rinse the grinder and tube separately, then poured into the same conical tube. The total amount of water used was 15 mL (0.5 mL/abdomen), allowing for a minimum detection level of 25,000 spores/bee. Samples were vortexed before being pipetted onto both sides of the hemocytometer to ensure an even distribution of spores. The samples were allowed to settle for 1 minute after being loaded into the hemocytometer, and both sides were counted [40]. After conversion to account for dilution, values were averaged to produce a unit of spores/bee. Samples that contained more than approximately 100 spores per square were further diluted to ensure accurate counting.

## Determining *Nosema* species

To determine which *Nosema* species were infecting the colonies, composite apiary samples from June 12–16, 2017, and April 24 and 26, 2018, were analyzed (8 samples total). All colonies from within the same apiary had *Nosema* samples of 30 bees frozen using liquid nitrogen and crushed in a mortar and pestle. An equal portion (measured in g) of the crushed sample from each colony was mixed to create an equivalent 30 bee sample. Crushed bee samples were stored at -80˚C until DNA extraction.

DNA was extracted from ~100 μL of homogenized bee using the DNeasy Blood and Tissue Kit (Qiagen, Mississauga, ON, Canada) with the QIAcube (Qiagen) automatic DNA extraction instrument along with its associated protocol. The lysing step with proteinase-K and Buffer ATL was done overnight using a thermomixer set to mix at 500 rpm for 15 s every 30 min. DNA concentration and purity were determined using a NanoDrop Lite spectrophotometer (Thermo Scientific, Wilmington, DE, USA). DNA samples were stored at -20˚C until further processing.

DNA samples were quantified for *N. apis* and *N. ceranae* using an Applied Biosystems Quantstudio 6 Flex (Thermo Scientific, Wilmington, DE, USA) qPCR instrument. Each 20 μL reaction contained 10 μL SsoFast Evagreen Supermix (Bio-Rad, Hercules, CA, USA), 1 μL primer (F + R), 5 μL DNA template, and 4 μL nuclease-free water. For *N. apis* and *N. ceranae*, the primers Na-321 [41] and Nc-104 [42] were used, respectively (both primers were obtained from Integrated DNA Technologies, Coralville, IA, USA). The primer Actin181 [43] was used for detecting bee actin (Integrated DNA Technologies). For each sample, the *Nosema* reactions were run in triplicate with a positive bee actin control. Each plate had a *Nosema* standard curve from $10^4$ to $10^8$ of synthesized target sequences (GBLOCK; Integrated DNA Technologies) as well as negative primer controls and no-template controls. All qPCR protocols were performed using under the following conditions: initial denaturation of 95˚C for 3 min, PCR cycling (40 cycles of 95˚C for 15s, 55˚C for 30s, and 72˚C for 30s), and melt curve analysis (raising the temperature from 65˚C to 95˚C in 0.5˚C increments with a hold of 5 seconds for each increment).

## Estimating adult bee population

The population of adult bees was estimated approximately once each month through the active beekeeping season (April-September). Depending on weather conditions, one of two methods was used to estimate bee population. The first method for measuring bee population during warm weather involved recording the percent (to the nearest 25%) of bees covering each surface (front and back) of every Langstroth frame (48 x 23 cm for one side) in the hive and multiplying by the known number of bees to cover a frame. It was assumed that 2430 bees fully cover both sides of a frame [44]. The second method for assessing the bee population was done by determining the colony cluster size to approximate bee population when inclement weather prevented a full assessment (late fall and early spring). This was done by viewing the top of the cluster from the top and bottom brood chambers and counting the number of frames covered by bees [45]. A cluster size estimate was used in April 2018, September 2018, and April 2019. Adult bee population was expressed as number of bees, whereas cluster size uses number of frames with bees. A colony was considered dead when no live queen or bees were left in the hive and recorded at the time of sampling. In the spring (April 2018, April 2019), colony viability was also assessed, with colonies being considered non-viable from a commercial standpoint when there were fewer than four frames of bees in the colony [46]. Queen presence, supersedure, and acceptance were recorded when observed throughout the study. When colonies became queenless, a new marked queen was introduced.

## Statistical analysis

The effects of sampling date, location, and wintering method on *Nosema* abundance, adult bee population, and cluster size were analyzed with PROC MIXED (SAS v.9.4) using a repeated measures design with colonies nested within apiary as the subject and sampling date as the repeated measure using the REML statement (restricted maximum likelihood). *Nosema* abundance was logarithmically transformed to meet the assumption of normality. The Kenward-Roger Degrees of Freedom Approximation was used to adjust for issues with homogeneity of variance. The following covariance structures were found to have the best fit; heterogeneous first-order autoregressive for *Nosema* abundance and first-order autoregressive for adult bee population. When significant interactions occurred, the Slice option in the LSMEANS statement was used to partition the effects by location and wintering method to compare the differences between means within date. Only live colonies were used in the adult bee population and cluster size analyses. Additionally, contrasts were used to find differences in *Nosema* abundance, adult bee population, and cluster size before and after winter by location and wintering method. Contrasts were also performed to determine if any changes over winter were different between locations or wintering methods. Analyses were performed on the transformed data but are presented as untransformed means.

A multivariate analysis of the effects of average *Nosema* abundance, location, and wintering method on colony mortality and non-viability after winter was performed using a binary logistic regression with backward elimination (PROC LOGISTIC, SAS v.9.4). Analyses were conducted for June 2017-April 2018 and April 2018–2019 as well as for the whole study (2017–2019). *Nosema* abundance was included in the analysis as either individual sampling dates or averages. Average *Nosema* abundance was calculated for the spring (April-June), summer (July-August), and fall (September), as well as the maximum *Nosema* spores/bee and average for the period leading up to winter. The whole study analyses included *Nosema* samples taken from dead bees (first sample after winter) that were collected separately. One colony was removed from the 2018–2019 logistic analysis due to having extremely high mite levels despite *Varroa* treatment. It should be noted that PROC LOGISTIC removes missing values for the response or explanatory variables from the analysis. This resulted in the removal of one dead colony from the first year and whole study analyses. The colony died early in the first year and would likely not change the results of the analyses. In the second year, only spring *Nosema* abundance (along with location and wintering method) was included in the analyses as most dead colonies died during the summer and fall.

## Results

The study began with 31 colonies; however, by autumn, only 29 colonies were alive (Fig 1). Six colonies (20.7%) died over winter 2017–2018, leaving 23 live colonies. Of these, an additional six colonies (26.1%) died during the summer of 2018. By winter 2018–2019, only 17 colonies were overwintered. One colony (5.9%) died over winter, leaving 16 of the 31 original colonies alive after 23 months.

All eight composite apiary bee samples that were analyzed using qPCR were infected with *Nosema*. In June 2017, all four apiaries were infected with only *N. ceranae*. In April 2018, three of the apiaries were infected with only *N. ceranae*, whereas one North apiary was infected with both *N. ceranae* and *N. apis*. In the mixed infection, *N. apis* had approximately four times the number of copies as *N. ceranae*.

### Epidemiology

*Nosema* abundance varied over time (F = 13.69, df = 19, 100, P < .0001). In general, *Nosema* abundance was low in the late summer and fall and highest in the spring. *Nosema* abundance

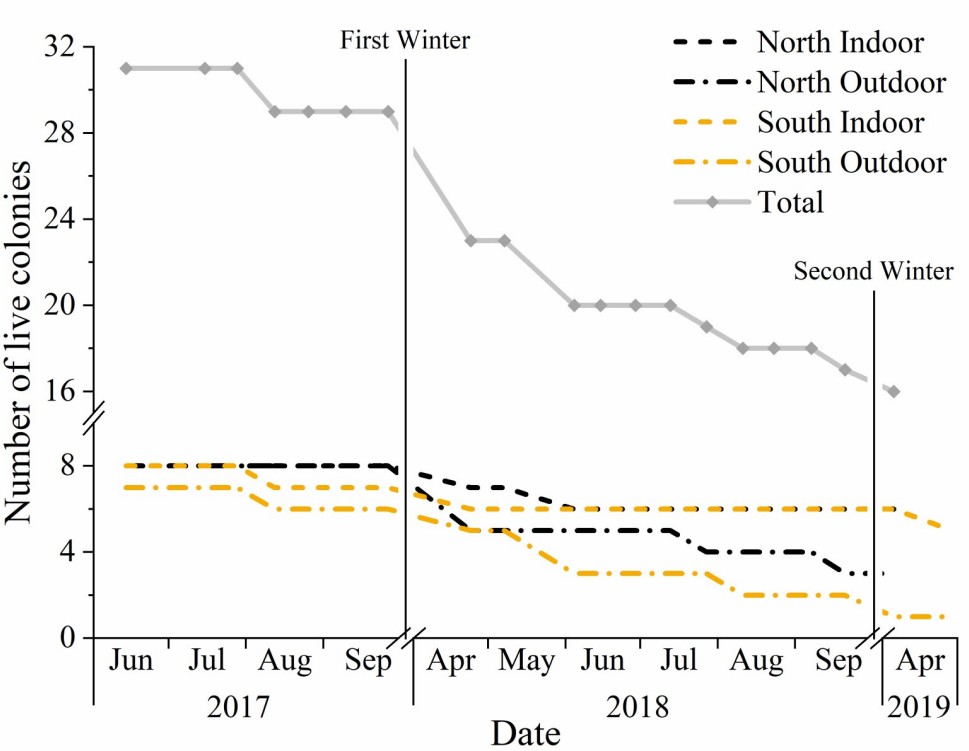

**Fig 1. Number of live colonies in total and by apiary (location and wintering method) over 23 months with x-axis breaks with vertical lines indicating winter.**

was significantly higher after winter than in samples taken before winter in September for both years (2017–2018: F = 28.85, df = 1, 43.6, P<0.0001; 2018–2019: F = 7.04, df = 1, 23.1, P = 0.014, Contrast). Furthermore, there was a significant interaction between location and date on *Nosema* abundance (F = 1.81, df = 18, 103, P = 0.033; Fig 2). The North colonies had significantly higher *Nosema* abundance than the South on July 28, 2017, and April 28, 2018 (P<0.05, Slice). Whereas the colonies in the North had lower *Nosema* abundance than the South on June 5 and 16, 2018 (P<0.05, Slice). *Nosema* abundance increased overwinter at both locations in 2017–2018 (P<0.05, Contrast). In 2018–2019, *Nosema* abundance increased over-winter only for North colonies (F = 8.95, df = 1, 23.8, P = 0.0064, Contrast), whereas the South colonies had similar levels before and after winter (F = 1.31, df = 1, 22.5, P = 0.26, Contrast). The relative change in *Nosema* abundance overwinter in North apiaries compared to South apiaries was not significantly different between locations for either winter (P>0.05, Contrast).

*Nosema* spore abundance did differ between wintering methods, but not consistently. After the first winter, outdoor-wintered colonies had a higher average *Nosema* abundance than indoor-wintered colonies, with the opposite occurring after the second winter (Fig 3). How-ever, the difference in either year was not significant (wintering method*sampling date: F = 0.04, df = 1, 100, P = 0.47). In 2017–2018, *Nosema* abundance increased over winter for both wintering methods (P<0.05, Contrast). In 2018–2019, *Nosema* abundance increased overwinter for indoor-wintered colonies (F = 4.86, df = 1, 23.8, P = 0.037, Contrast) but did not change significantly for outdoor-wintered colonies (F = 3.20, df = 1, 22.5, P = 0.087, Con-trast). The relative change in *Nosema* abundance over winter was not significantly different between wintering methods for either winter (P>0.05, Contrast).

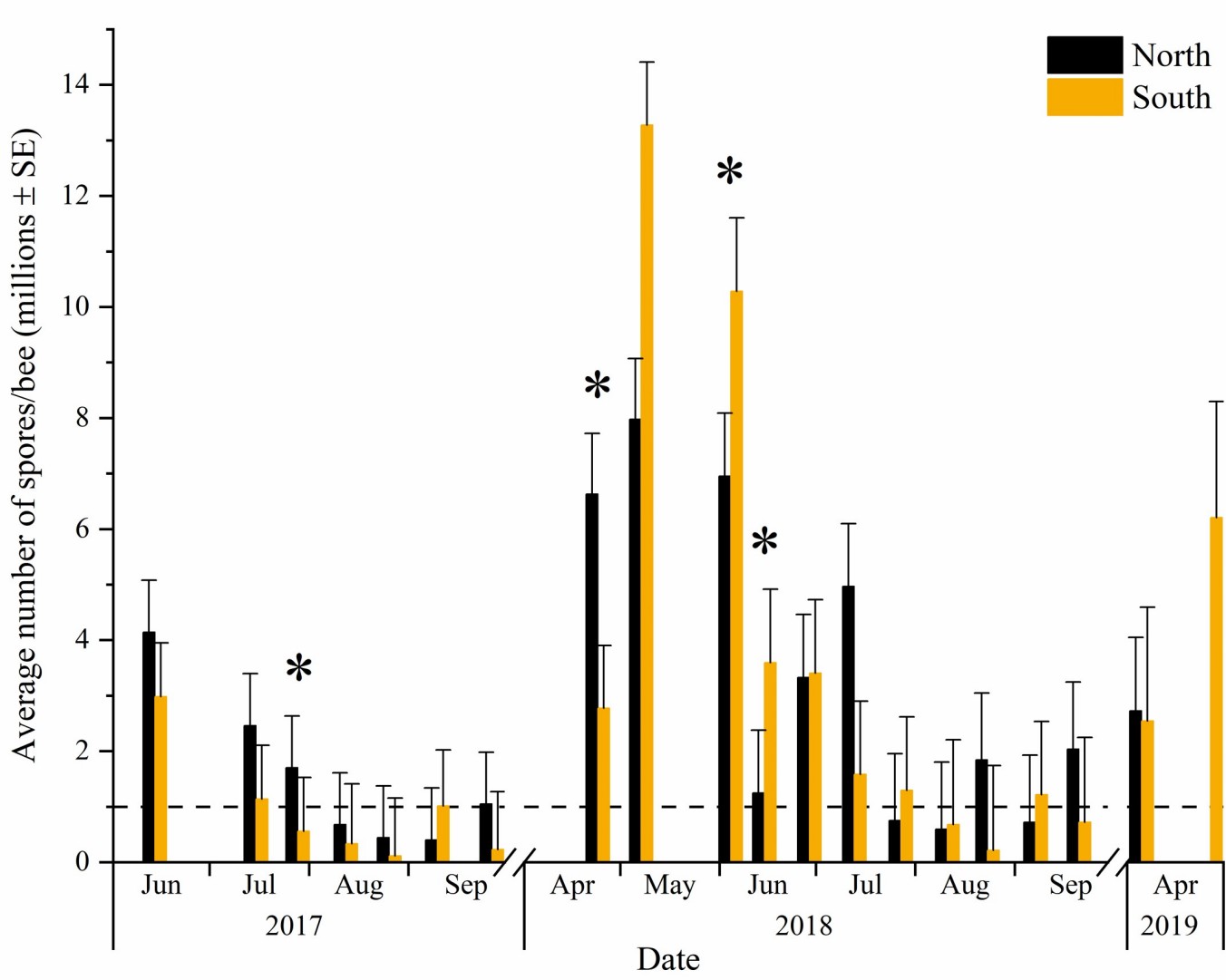

**Fig 2. Influence of geographic location on average *Nosema* abundance over 23 months.** North represents colonies near Edmonton and South represents colonies near Rainier. Data are plotted as untransformed means. Dashed line shows the 1 million spores/bee nominal threshold. Asterisks indicate significant differences (P <0.05, Slice) between locations within dates.

### Effect on bee population

In this study, location did not affect adult bee population (live colonies) over time (F = 1.07, df = 8, 153, P = 0.38). In contrast, there was a significant interaction among wintering method and sampling date on adult bee population (F = 2.01, df = 8, 153, P = 0.048). Indoor-wintered colonies built up larger bee populations than outdoor-wintered colonies during the subsequent period of population growth in spring and summer, from June to July 2018 (P<0.05, Slice; Fig 4). Over winter 2017–2018, adult bee population was significantly lower after winter than before winter (F = 123.72, df = 1, 158, P<0.0001, Contrast). The change in adult bee population over winter was not significantly different between locations or wintering methods (P>0.05, Contrast). Neither location nor wintering method affected cluster size (number of frames with bees) in September 2018-April 2019 (F = 0.33, df = 1, 11.8, P = 0.58, and F = 1.85, df = 1, 12.4, P = 0.20, respectively). Cluster size decreased overwinter 2018–2019 in both

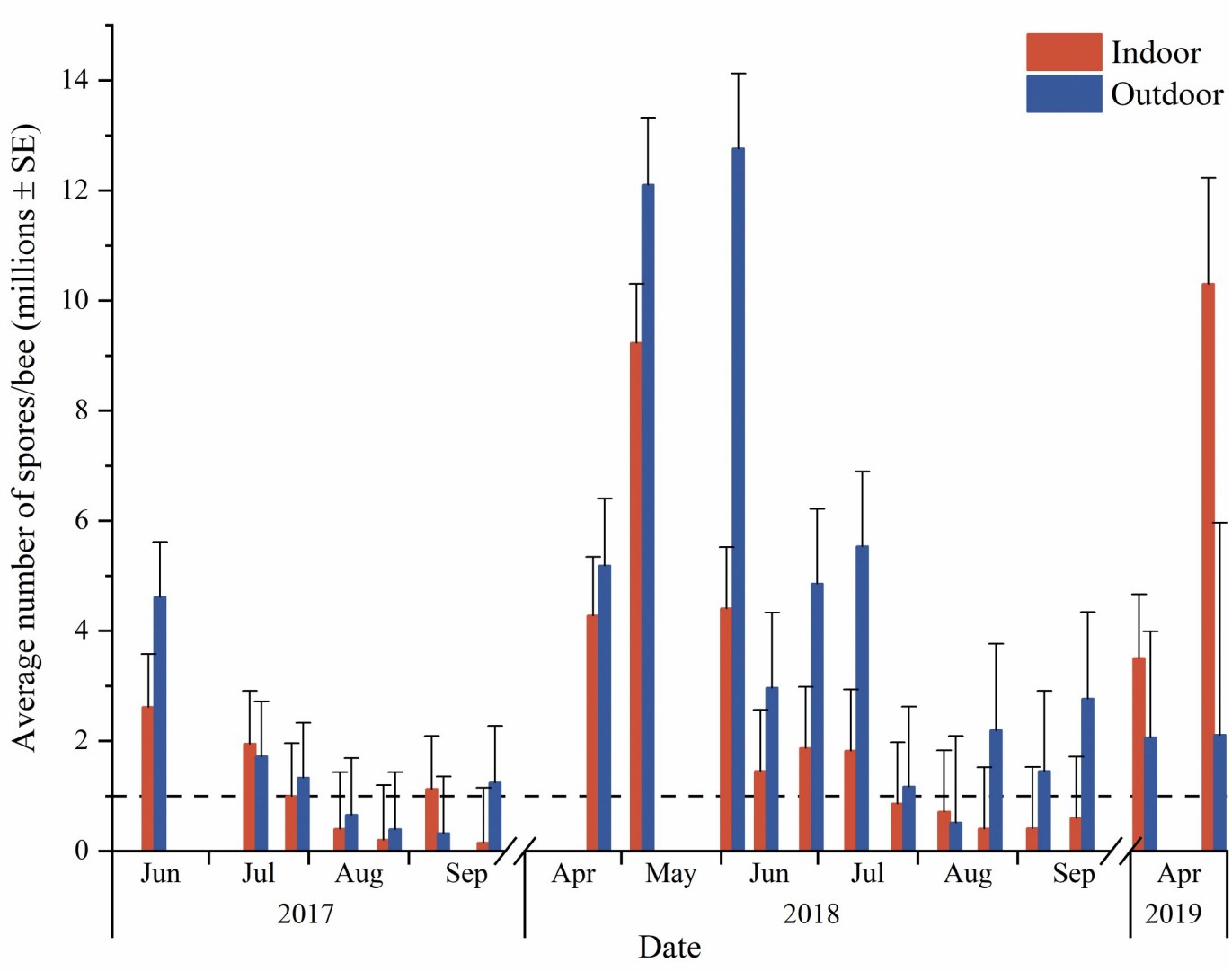

**Fig 3. Influence of wintering method on average *Nosema* abundance over 23 months.** Indoor represents colonies moved to an indoor wintering building, and outdoor represents colonies kept outdoors in insulating wraps. Data are plotted as untransformed means. Dashed line shows the historical 1 million spores/bee nominal threshold.

locations (P<0.05, Slice). Over the winter of 2018–2019, the cluster size of surviving colonies decreased for indoor-wintered colonies (F = 27.36, df = 1, 11.2, P = 0.0003, Slice), whereas outdoor-wintered colonies that survived had similar cluster sizes before and after winter (F = 2.01, df = 1, 11.8, P = 0.18, Slice). The change in cluster size overwinter in 2017–2018 did not differ among locations or wintering methods (P>0.05, Contrast).

## Predicted mortality

Multivariate analyses showed that colonies with higher *Nosema* abundance in summer 2017 (July-August) were more likely to die by the following spring (April 25, 2018) ($\chi^2$ = 3.9486, df = 1, P = 0.047; Fig 5A). When the average summer *Nosema* abundance was lower than the one million spores/bee threshold, there was less than a 20.5% chance of mortality. In the second year, higher spring 2018 (April-June) *Nosema* abundance significantly increased the

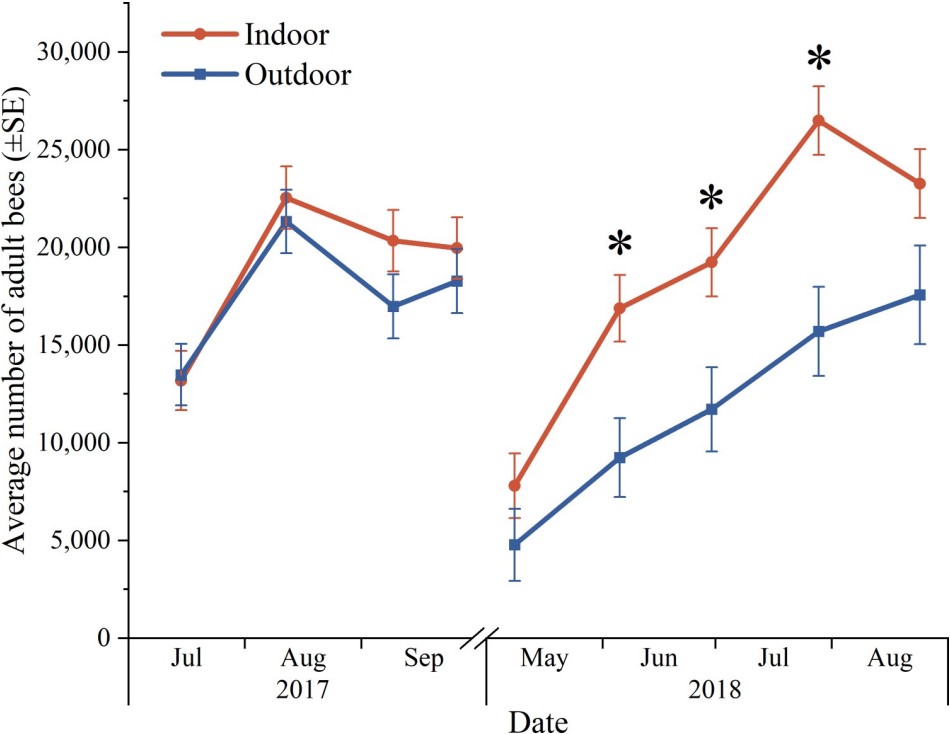

**Fig 4. Influence of wintering method on average adult bee population (live colonies) over 23 months.** Indoor represents colonies moved to an indoor wintering building, and outdoor represents colonies kept outdoors in insulating wraps. Asterisks indicate significant differences between wintering methods.

probability of mortality in the following spring (April 5, 2019) ($\chi^2$ = 4.8374, df = 1 P = 0.028; Fig 5B). There was less than a 3.1% chance of mortality when the average spring *Nosema* abundance below the one million spores/bee threshold.

There was a significant effect of *Nosema* spore level on overall colony mortality over the almost two-year study. Over both years, colonies with a higher two-year average *Nosema* abundance (June 2017-September 2018) were more likely to die by the end of the second winter ($\chi^2$ = 4.9830, df = 1, P = 0.026; Fig 6). Additionally, outdoor-wintered colonies were more likely to die than indoor-wintered colonies ($\chi^2$ = 4.9896, df = 1, P = 0.026; Fig 6). There was no significant interaction between *Nosema* abundance and wintering method, indicating that indoor-wintered colonies had increased survival relative to outdoor-wintered colonies across the range of spore levels found in this study. More specifically, colonies with a higher two-year average *Nosema* abundance in the spring (June 2017, April-June 2018) were more likely to die by the end of the study ($\chi^2$ = 4.8524, df = 1, P = 0.028; Fig 7). There was less than a 16.7% chance of mortality by the end of the study when the two-year average spring *Nosema* abundance was below the one million spores/bee threshold. The probability of having non-viable surviving colonies in the spring was not predicted by *Nosema* abundance, location, or wintering method.

## Discussion

This study demonstrates a distinct seasonal pattern of *N. ceranae* abundance in untreated colonies under Canadian Prairie conditions, specifically in Alberta, with high spore loads in spring declining to low levels in the summer. Unlike traditional patterns of *N. apis*, an increase in the

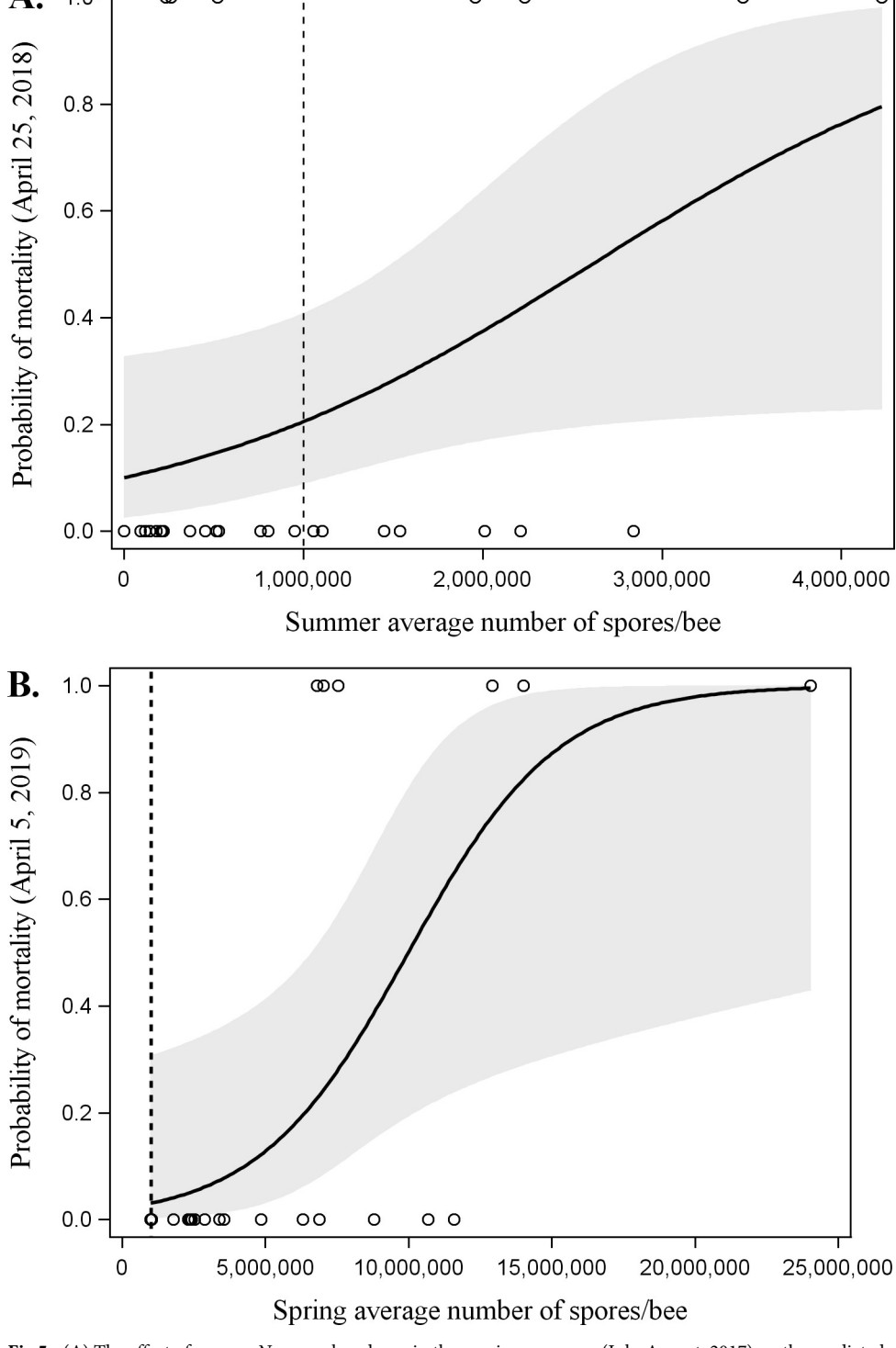

**Fig 5.** (A) The effect of average *Nosema* abundance in the previous summer (July-August, 2017) on the predicted probability of having dead colonies in spring 2018 (cumulative colony mortality on April 25, 2018). (B) The effect of average *Nosema* abundance in the previous spring (April—June, 2018) on the predicted probability of having dead colonies in spring 2019 (cumulative colony mortality on April 5, 2019). Shaded area is the 95% confidence limit. Dashed line shows the 1 million spores/bee nominal threshold. The circles indicate whether colonies were alive (0) or dead (1) at a specific *Nosema* abundance.

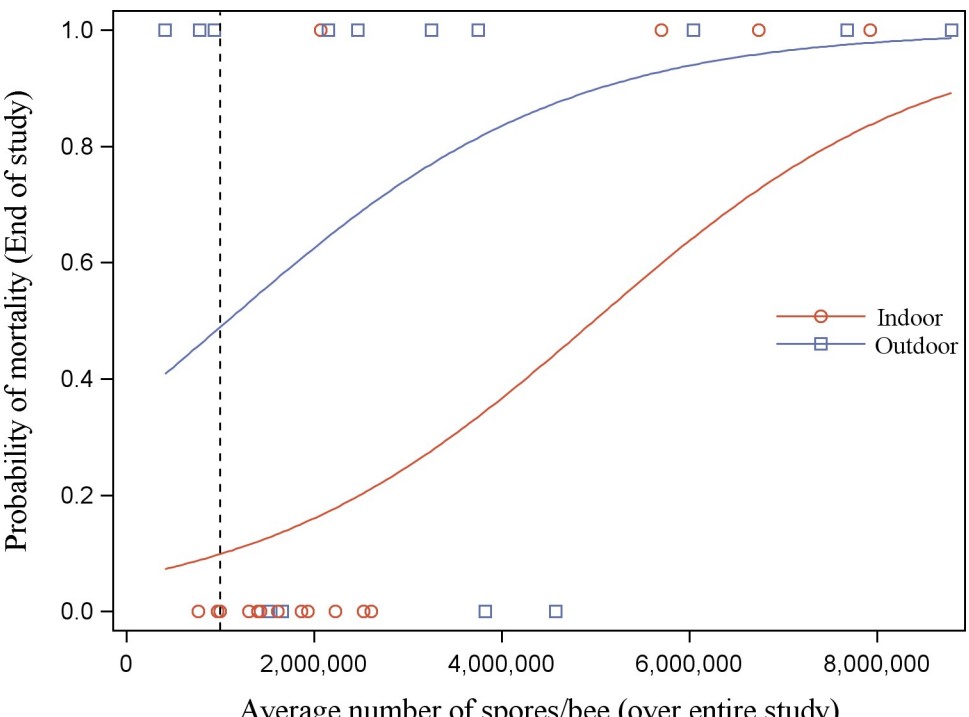

**Fig 6. The effect of wintering method on the predicted probability of observing dead colonies at the end of 23 months (cumulative mortality at the end of the study) as influenced by the average *Nosema* abundance over the study (June 2017-September 2018).** Indoor represents colonies moved to an indoor wintering building, and outdoor represents colonies kept outdoors in insulating wraps. Dashed line shows the 1 million spores/bee nominal threshold. The symbols indicate whether colonies were alive (0) or dead (1) at a specific *Nosema* abundance within wintering method. Samples of dead bees taken from the bottom board (due to the unavailability of live bees in the colony) are included in the average.

fall period when beekeepers usually monitor for *Nosema* was not observed for *N. ceranae* in this study. This has implications for managing this pathogen as fall assessments may not be the best indicator of future colony health outcomes, *Nosema*-induced stress, or treatment requirements. There was no consistent pattern for differences in *Nosema* abundance by location, possibly due to the highly variable nature of this pathogen or the small number of sites. Similarly, wintering method did not consistently affect *Nosema* abundance following winter. However, colonies that were wintered indoors had lower mortality and faster spring population build-up than outdoor-wintered colonies. Also, when averaged over the two-year study, *Nosema* abundance in the spring was a significant predictor of end-of-study colony mortality, highlighting the importance of spring *Nosema* monitoring and treatments.

In this study, apiaries were either infected predominately with *N. ceranae* or, to a lesser degree, co-infected with *N. ceranae* and *N. apis*. This represents an increasing trend of *N. ceranae* becoming the dominant form of this pathogen in Canada. In 2010, 41% of Alberta colonies had single *N. ceranae* infections, and 25% had infections containing both species [10]. Recent studies in Ontario, Quebec, and Maritime Provinces also show *N. ceranae* as the predominant species [10–14]. While this study did not include only pure infections of *N. ceranae* throughout the study, it is reasonable to assume that *N. ceranae* is the species causing the seasonal patterns and impact on colonies observed in this study, given its numerical dominance.

Although several studies in Europe showed unpredictable seasonal patterns for *N. ceranae* [4,21,41], our results from 4 sites over two years showed a consistent spring peak and low

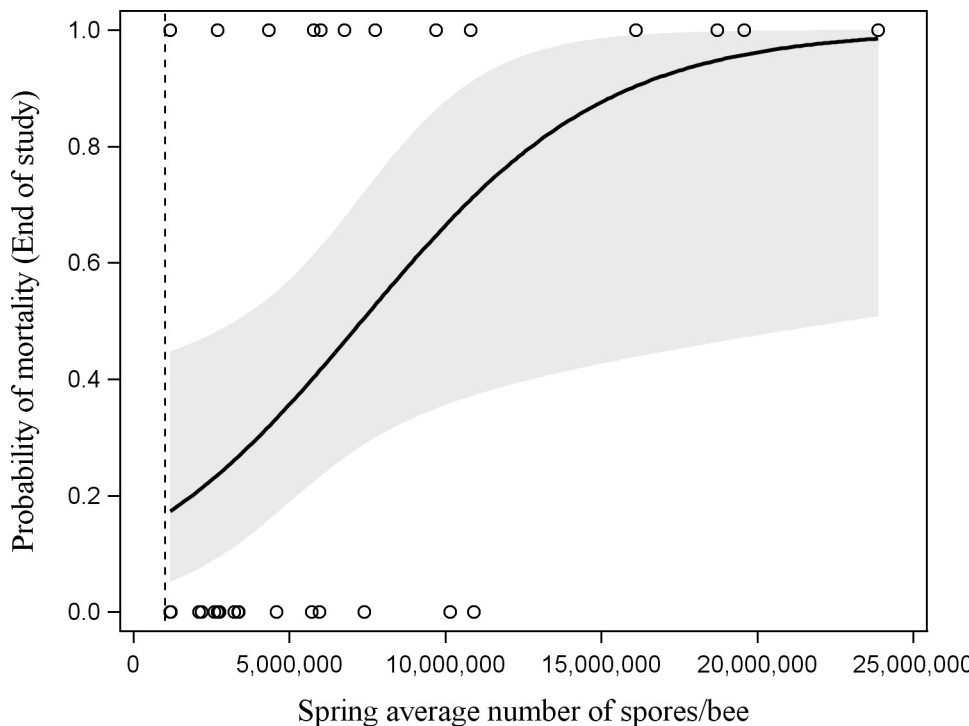

**Fig 7. The effect of two-year average spring *Nosema* abundance over the study (June 2017, April-June 2018) on the predicted probability of observing dead colonies at the end of 23 months (cumulative mortality at the end of the study).** Shaded area is the 95% confidence limit. Dashed line shows the 1 million spores/bee nominal threshold. The circles indicate whether colonies were alive (0) or dead (1) at a specific *Nosema* abundance. Samples of dead bees taken from the bottom board (due to the unavailability of live bees in the colony) are included in the average.

summer and fall levels, supporting the findings of previous North American studies [27,30,31,47]. In contrast, a 3-year study in Quebec, Canada, found *N. ceranae* infection peaks were at a different time for each year of the study, including the fall, summer, and spring [11]. High variation in their study could be due, in part, to the study's small sample size (8 colonies total). The seasonal pattern for *N. ceranae* seen in this study has some similarities to the established pattern for *N. apis*, which has a spring peak and low summer levels, but unlike *N. ceranae*, typically shows a prominent fall peak [11,28,29]. It should be noted that it is possible that our sampling would have missed a potential late fall peak as sampling ended in September in both years. A recent study from Nova Scotia found *N. ceranae* had a small peak in October though it did not exceed the 1 million spores/bee threshold [31]. Typically, beekeepers monitor for *Nosema* in the early fall (September) to determine if treatment is needed. However, if the fall peak is detected in October based on the current threshold, it would be too late to treat for the coming winter under Alberta conditions. By this point, colonies are being winterized due to the rise of freezing temperatures. Additionally, bees do not take syrup as readily at these temperatures, making treatment impossible. The difference in fall *Nosema* abundance for these two species further demonstrates that sampling timing when using the nominal one million spores/bee threshold needs to be reassessed in order to make sound control decisions and prevent economic losses.

This study found there was no consistent pattern for differences in *Nosema* abundance by location within Alberta. It was predicted that *Nosema* abundance would be lower in the South than in the North, where the average annual temperature is higher by 2.1˚C and colonies are often exposed to warm periods in winter that allow for defecation flights (Table B in S1 File).

During *Nosema* sampling, the South was consistently warmer than the North (Table C in S1 File). However, at most sampling dates, there was no difference in *Nosema* abundance between the two locations. When differences did occur, sometimes the North had higher *Nosema*, and sometimes it was the South. Additionally, *Nosema* abundance increased in both locations over-winter in 2017–2018, but in 2018–2019, only colonies in the North saw an increase in *Nosema* over winter. The winters in the South varied substantially between the two years, and abnormally low temperatures late in the first winter may explain the relatively high winter *Nosema* abundance. March 2018 had no days above 10°C, whereas March 2019 had 11 days, which is more consistent with the 20-year average (Table B, C in S1 File). Although bees performing cleansing flights during warm periods near the end of the second winter may have allowed them to reduce their spore load, it should be noted that the change in *Nosema* abundance over-winter was not different between the North and South for either winter. Also, the small number of sites in this study may not adequately represent Alberta's full climatic variation.

In this study, the effects of wintering method on *Nosema* abundance in fumagillin-free colonies sampled after winter were not consistent. After the first winter, outdoor-wintered colonies had numerically greater *Nosema* abundance than indoor-wintered colonies. In contrast, *Nosema* abundance increased over the second winter for indoor-wintered colonies, while outdoor-wintered colonies had similar abundance before and after winter. It appears that factors from before winter caused differences in wintering treatment. A similar lack of consistency was seen in Nova Scotia where overall there was no effect of wintering method on *Nosema* abundance in fumagillin-treated colonies, except for one beekeeping operation where outdoor wintered colonies had higher abundance than indoor-wintered colonies [13]. In Manitoba, *Nosema* abundance decreased in indoor-wintered colonies but increased in outdoor colonies [25]. It should be noted that both treated and untreated colonies were included in their analysis.

However, wintering method had a significant impact on colony performance related to *Nosema* infection level. The data suggests that indoor-wintered colonies were less likely to die from *Nosema* infections and have larger bee populations in the following summer than outdoor-wintered colonies. However, our data did not conclusively show this and must be clarified through further studies that manipulate *Nosema* levels in indoor and outdoor winter environments are needed. Our study supports other Canadian studies that have demonstrated that colonies infected with high levels of parasites and/or pathogens were more likely to survive when wintered indoors than when wintered outdoors [13,25,48]. The apparent difference in colony mortality between wintering methods across all levels of *Nosema* abundance suggests that the threshold for damage tolerated by beekeepers from *Nosema* infection should be lower for colonies that will be wintered outdoors to have survival similar to indoor-wintered colonies, which also needs to be confirmed with further studies. The reduced population growth of outdoor-wintered colonies may also be due to *Nosema*. Although not significantly different, *Nosema* abundance in outdoor-wintered colonies trended higher than indoor-wintered colonies from May to mid-July. *Nosema ceranae* infection has been shown to reduce colony population growth in other studies [21,22,27,49]. These results showed that indoor wintering is an attractive option for beekeepers looking to reduce mortality and increase summer colony population size.

It was predicted that temperature stress and cleansing flight opportunities could impact *Nosema* abundance, but the relative importance of these factors was unknown. Indoor-wintered colonies experience lower temperature stress and no cleansing flights, whereas outdoor-wintered colonies would have greater temperature stress with cleansing flight opportunities varying with the local climate. This study found that colonies performed better when wintered indoors than wintered outdoors. These results suggest that mitigation of temperature stress

during indoor wintering may have affected the impact of *Nosema* infections on colonies more than any benefits associated with the availability of late winter cleansing flights. This may be due to cleansing flight opportunities being unpredictable and short-lived, while reduced temperature stress can occur all winter. These results do not preclude the possibility that other factors, such as variation in relative humidity in the different environments or mid-winter brood-rearing levels, could have also affected the results.

*Nosema* abundance was found to be a significant predictor of colony mortality. At the end of the first year (April 25, 2018), the probability of colony mortality was predicted by summer *Nosema* abundance (July—August 2017). In contrast, spring *Nosema* abundance (April-June 2018) predicted colony mortality at the end of the second year (April 5, 2019). The difference between the first and second years may be due to missing the spring *Nosema* peak in 2017. The spring peak in the second year of this study occurred in April to May, but spring 2017 included only one *Nosema* sample taken in June. However, this appears to be accounted for by taking the two-year average as *Nosema* abundance in the spring (June 2017, April-June 2018) was a significant predictor of end-of-study colony mortality. While previous Canadian studies have found that *N. ceranae* was not correlated with mortality [13,25–28], these studies restricted sampling to the fall through to the spring as it was assumed that fall *Nosema* abundance were most likely to predict winter mortality. Perhaps these studies would have found associations between *Nosema* and mortality if sampling had been carried out in the spring to early summer. Our study has shown that spring and summer *Nosema* abundance is a better predictor of mortality when colonies are not treated than early fall abundance. Coupled with the lack of a fall peak, this suggests that spring fumagillin treatments may be more important than fall fumagillin treatments in managing this species of *Nosema* in Alberta.

In conclusion, this study demonstrates the presence of a seasonal pattern of *N. ceranae* abundance in the Canadian Prairies, with higher *Nosema* abundance in the spring than in summer or early fall. This study found no consistent evidence that *Nosema* is a greater threat to colony health in the colder 'North' region of Alberta than the relatively warmer South. There was no consistent effect of wintering method on *Nosema* abundance; however, indoor wintering was more effective in increasing colony survival than outdoor wintering under the range of *Nosema* abundances found in this study. Also, indoor-wintered colonies had greater populations in the following spring than their outdoor-wintered counterparts. Therefore, to achieve similar mortality to indoor-wintered colonies, the data suggests that the one million spores/bee threshold as assessed in spring needs to be lowered for colonies being wintered outdoors. However, this requires further research. This study found clear evidence that spring *N. ceranae* impacts the health of the colony, as the probability of survival decreased with increasing *Nosema* loads above 1 million spores/bee. Further research is needed to determine appropriate seasonal thresholds for *N. ceranae* and how wintering method interacts with pests and disease to affect colony health and survival. It also needs to be determined if beekeepers could benefit more from treating *Nosema* in the spring than in the fall, as our results suggest it may have a greater impact in reducing colony mortality.

## Supporting information

**S1 File.** Supporting information for: Table A. Timeline for honey bee sampling and bee population estimates from May 2017 to April 2019 in Alberta colonies; Table B. Monthly climate normals data from weather stations near apiary sites; Table C. Monthly climate data from weather stations near apiary sites for 2017–2019.
(DOCX)

**S2 File.** Data used for: Dataset A. Raw data used for PROC MIXED analysis; Dataset B. Raw data used for PROC LOGISTIC analysis including samples with dead bees taken from bottom board; Dataset C. Raw data used for PROC LOGISTIC analysis not including samples with dead bees taken from bottom board.
(XLSX)

## Acknowledgments

We thank Rassol Bahreini for his advice in the early planning of this project. We thank the Government of Alberta Bee Health Assurance Section–Bee Teams who assisted with sample collection and processing. We thank Zoe Rempel for providing technical support for the qPCR analysis. We thank Sawyer Patterson for analyzing brood data. We wish to thank the Alberta beekeepers who provided honey bee colonies for this study: Scandia Honey Corporation and Apiaries of Alberta Pride.

## Author Contributions

**Conceptualization:** Rosanna N. Punko, Robert W. Currie, Medhat E. Nasr.

**Formal analysis:** Rosanna N. Punko, Robert W. Currie.

**Funding acquisition:** Robert W. Currie, Medhat E. Nasr, Shelley E. Hoover.

**Investigation:** Rosanna N. Punko.

**Methodology:** Rosanna N. Punko, Robert W. Currie, Medhat E. Nasr.

**Resources:** Robert W. Currie, Medhat E. Nasr.

**Supervision:** Shelley E. Hoover.

**Visualization:** Rosanna N. Punko.

**Writing – original draft:** Rosanna N. Punko.

**Writing – review & editing:** Rosanna N. Punko, Robert W. Currie, Medhat E. Nasr, Shelley E. Hoover.

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
