## [Decision Letter · Decision Letter 0]

2 Aug 2021

PONE-D-21-14529

Epidemiology of *Nosema* spp. and its effect on colony size and survival under indoor and outdoor wintering in the Canadian Prairies

PLOS ONE

Dear Dr. Punko,

Thank you for submitting your manuscript to PLOS ONE. After careful consideration, we feel that it has merit but does not fully meet PLOS ONE’s publication criteria as it currently stands. Therefore, we invite you to submit a revised version of the manuscript that addresses the few minor points raised during the review process (for details, please see below).

We look forward to receiving your revised manuscript.

Kind regards,

Wolfgang Blenau

Academic Editor

PLOS ONE

Additional Editor Comments (if provided):

Reviewers' comments:

Reviewer's Responses to Questions

**Comments to the Author**

1. Is the manuscript technically sound, and do the data support the conclusions?

Reviewer #1: Partly

Reviewer #2: Yes

2. Has the statistical analysis been performed appropriately and rigorously? 

Reviewer #1: Yes

Reviewer #2: Yes

3. Have the authors made all data underlying the findings in their manuscript fully available?

Reviewer #1: Yes

Reviewer #2: Yes

4. Is the manuscript presented in an intelligible fashion and written in standard English?

Reviewer #1: Yes

Reviewer #2: Yes

5. Review Comments to the Author

Reviewer #1: General comment:

I enjoyed reading this article that reports results on the epidemiology of Nosema ceranae and the impact of indoor wintering of honey bee colonies in the Canadian prairies. I know studies like this one require considerable amount of work and coordination, and thus I commend the authors for their efforts. The authors found that the intensity of N. ceranae infections increased during spring and decreased the rest of the seasons. They also found that N. ceranae infections in spring and summer were good predictors of overwinter colony mortality (one of the most important results of this study). Additionally, they demonstrated that indoor wintering results in higher survivorship and larger colony populations after winter, compared to overwintering colonies outdoors. However, I do not think the data support the conclusion that indoor wintering reduced the impact of N. ceranae infections on colony mortality or development. I find this report valuable, and I have some comments and suggestions aimed at improving the ms.

Specific comments to the authors:

1. Title. I suggest rewording it. The data do not support the conclusion that indoor wintering reduced the impact of N. ceranae infections on colony mortality or development. Something like: “Epidemiology of Nosema spp. and effect of indoor and outdoor wintering on survival and populations of honey bee colonies in the Canadian prairies.” Or other similar title, not implying an interaction between wintering system and Nosema spp. infections on colony survival and growth.

2. Abstract, line 2. Could write Nosema ceranae or Nosema spp.

3. Abstract, lines 7 and 8. Could you please specify what is meant by “economic viability”? The results do not show an economic analysis. Otherwise, delete it.

4. Abstract, lines 14-17. What is stated in those lines are a recommendation and a conclusion not supported by the data. Should be reworded or deleted.

5. Introduction, line 96: “economic viability” Please see comment 3 above.

6. Methods. Excellent experimental arrangement and setup of colonies.

7. Methods, lines 152-166. Good description of methods to determine varroa infestation levels. However, as a reader, I expected to see results on these measurements but they are not reported. Therefore, I suggest to either report and discuss those results, or just mention that you controlled varroa infestations by treating all colonies in the fall, deleting the text related to monitoring varroa infestations.

8. Results. Perhaps rearrange this section with subheadings to report results in different subsections (epidemiology, mortality prediction, effect of wintering method, etc.). That will help the reader to better understand and interpret the results.

9. Results. Words like “slice” or “contrast” are inserted in the parentheses reporting probabilities. Why is that? I had never seen such a thing. So, please delete or explain.

10. Results, lines 304-313. If no significant effects were found for Nosema spore abundance for either method or between methods in any year, the initial statement: “Wintering method had effects on Nosema spore abundance” is not accurate and should be reworded.

11. Discussion. Good discussion, but the main issue is to try to infer that wintering method interacted with N. ceranae infections, which resulted in differential survival or population growth of colonies. Your experimental design and statistical methods do not allow reaching that conclusion. It could be said that this is a possible explanation of the results (as a hypothesis rather than as a conclusion), but that further research is needed to separate the effects of wintering method and Nosema spp. infections or demonstrate their interaction on overwinter survival and posterior colony growth. The difference in colony survivorship and development found in this study could be mainly due to the wintering method since you could not demonstrate effects of wintering method on Nosema spp. abundance. You also do not show a correlation between Nosema abundance in winter and mortality or colony development. And no interaction effects between Nosema spp. abundance and wintering method on colony survivorship and growth are shown. Experiments with proper controls (Nosema-free colonies and Nosema-positive colonies treated with fumagillin) would be required in future experiments. Because of the above, I suggest rewording the parts of the discussion where it is inferred that there is a link between wintering method and Nosema spp. infections on colony survivorship and growth. As suggested before, reword it as a potential explanation, not as a conclusion.

12. Figures 5-8 could be Fig 5 a,b,c,d. Figure 8: line patterns are not clearly different.

Reviewer #2: In this article the authors were whether Nosema abundance differed between northern and southern locations in Alberta, whether nosema abundance differed in outdoor vs indoor wintered colonies. They followed up with examining bee population buildup, followed colony survival, and economic viability over the almost 2 year period. The finding that N. apis had 4X higher copies than N. ceranae is interesting. In other studies with mixed infections, N. apis is usually at much lower levels – maybe an effect of latitude?

General comments:

The paper was well written and easy to follow with nice results. The authors are familiar with the literature and presented their findings in context with other studies well.

Where does the 1 million spore/bee threshold come from? Is this only for N. apis? I am not aware that one had been established for N. ceranae but rather used the same assumption of the N. apis threshold. This may not be appropriate given the differences in impacts on bees by the two species.

Overall the figures are well presented and easy to follow. Figures 7 and 8 could be clarified more. The square and circles at the top and bottom do not seem to mean much as presented. What do the dashed lines represent? It is difficult to determine which is indoor vs outdoor.

Treatment would be best before the levels peak in the spring, however there are issues with residuals and harvesting the honey.

The economic viability aspect was not touched on as much. With “economic viability” I think that I am going to see numeric comparisons. How expensive is it to overwinter colonies? How does that cost compare to the difference in mortality from indoor vs outdoor overwintered colonies? Maybe a recommendation would be to not winter outdoors after the first winter but after the second where there is a significant difference in the number of adult bees.

Specific comments:

Line 81: I have limited knowledge on overwintering bees indoors. Are bees allowed to take “scheduled” defecation flights while indoor?

Lines 140-141: Was there a reason for using queens from Kona Queen? Why not use locally adapted queens?

Lines 419-420: why is treating with fumagillin in October too late in Alberta? Is it because the temperatures are already freezing? Would it be feasible if the colonies were overwintered indoors where there could be a controlled increase in temperature and light to allow them to break cluster to feed?

6. PLOS authors have the option to publish the peer review history of their article (what does this mean?). If published, this will include your full peer review and any attached files.

Reviewer #1: No

Reviewer #2: No

---

## [Author Response · Author response to Decision Letter 0]

30 Sep 2021

We thank the editor and two reviewers for their constructive comments on our manuscript and for the opportunity to revise and resubmit. Below is our response to each point raised by the editor and reviewers. All line references refer to those in the revised manuscript with tracked changes.

Responses to the Editor:

 Author Response: At line 238, ‘analysis’ was uncapitalized in the subheading. A space was also added between the last paragraph and the following headings: Acknowledgements and References. We hope that the manuscript fits the style requirements as described in the referred templates.

 Author Response: No permits were required for the study as the colonies used in this study were owned by commercial beekeeper cooperators. The field sites and wintering buildings were used with permission from the beekeepers. Their company names have been added to the acknowledgements section.

 Author Response: The grant number for the Alberta Crop Industry Development Fund (ACIDF) is #2014COO4R. University of Manitoba. Growing Forward 2 was a short-term government funding program that has since ended so grant numbers are no longer accessible. The University of Manitoba, Alberta Beekeepers Commission, and Canadian Bee Research Fund do not assign internal grant numbers. We are more than willing to provide any additional clarification if needed.

Responses to Reviewer #1: 

General comment:

I enjoyed reading this article that reports results on the epidemiology of Nosema ceranae and the impact of indoor wintering of honey bee colonies in the Canadian prairies. I know studies like this one require considerable amount of work and coordination, and thus I commend the authors for their efforts. The authors found that the intensity of N. ceranae infections increased during spring and decreased the rest of the seasons. They also found that N. ceranae infections in spring and summer were good predictors of overwinter colony mortality (one of the most important results of this study). Additionally, they demonstrated that indoor wintering results in higher survivorship and larger colony populations after winter, compared to overwintering colonies outdoors. However, I do not think the data support the conclusion that indoor wintering reduced the impact of N. ceranae infections on colony mortality or development. I find this report valuable, and I have some comments and suggestions aimed at improving the ms.

 Author Response: Thank you for your comments and compliments. We agree and have made the necessary changes to the manuscript to reflect that we did not show a direct impact of wintering method on Nosema abundance and colony mortality during winter. See comment 11 for the response to the data not supporting the conclusion of an interaction between wintering method and N. ceranae on survival and colony development. 

Specific comments:

1. Title. I suggest rewording it. The data do not support the conclusion that indoor wintering reduced the impact of N. ceranae infections on colony mortality or development. Something like: “Epidemiology of Nosema spp. and effect of indoor and outdoor wintering on survival and populations of honey bee colonies in the Canadian prairies.” Or other similar title, not implying an interaction between wintering system and Nosema spp. infections on colony survival and growth.

 Author Response: The title has been changed to “Epidemiology of Nosema spp. and the effect of indoor and outdoor wintering on honey bee colony population and survival in the Canadian Prairies” as suggested by the reviewer.

2. Abstract, line 2. Could write Nosema ceranae or Nosema spp.

 Author Response: Changed to Nosema spp. (line 2).

3. Abstract, lines 7 and 8. Could you please specify what is meant by “economic viability”? The results do not show an economic analysis. Otherwise, delete it.

 Author Response: The term ‘economic viability’ is misleading as we did not perform any economic analysis. However, we want to differentiate weak colonies from strong colonies coming out of winter. In short, a colony that has a small number of frames of bees will likely not produce honey that year and is considered by beekeepers ‘as good as dead.’ The Canadian Association of Professional Apiculturist has defined viable colonies in a commercial setting as having a minimum of 4 frames of bees following winter (line 233-235). Therefore, the term ‘economic viability’ has been changed to ‘commercial viability’ at line 8 and 99.

4. Abstract, lines 14-17. What is stated in those lines are a recommendation and a conclusion not supported by the data. Should be reworded or deleted.

 Author Response: Agreed. The text at lines 15-16 now reads as follows:

 “The results suggest that the existing Nosema threshold should be reinvestigated with wintering method in mind to provide more favorable outcomes for beekeepers.”

5. Introduction, line 96: “economic viability” Please see comment 3 above.

 Author Response: Made correction, see response under comment 3.

6. Methods. Excellent experimental arrangement and setup of colonies.

 Author Response: Thank you.

7. Methods, lines 152-166. Good description of methods to determine varroa infestation levels. However, as a reader, I expected to see results on these measurements but they are not reported. Therefore, I suggest to either report and discuss those results, or just mention that you controlled varroa infestations by treating all colonies in the fall, deleting the text related to monitoring varroa infestations.

 Author Response: Agreed. The Varroa section has been removed and it is noted under the experimental design that Varroa was controlled in the study. The revised text removed lines 157-167 and reads as follows:

 “Varroa mite populations were monitored throughout the study and maintained below the economic threshold (3%). Varroa was controlled in all colonies with Apivar® (500 mg Amitraz/strip) at the beginning of September in 2017 and 2018 as some colonies had infestation levels above the 3 mites per 100 bees fall threshold [38].”

8. Results. Perhaps rearrange this section with subheadings to report results in different subsections (epidemiology, mortality prediction, effect of wintering method, etc.). That will help the reader to better understand and interpret the results.

 Author Response: The following subheadings have been added to the results section- Epidemiology (line 289), Effect on bee population (line 326), and Predicted mortality (line 349).

9. Results. Words like “slice” or “contrast” are inserted in the parentheses reporting probabilities. Why is that? I had never seen such a thing. So, please delete or explain.

 Author Response: Many journals require that pre-planned or post-hoc tests be reported in the text following the statistical results. In SAS, the ‘slice’ command is a single degree-of-freedom contrast, equivalent to a protected LSD. The ‘contrast’ command is an orthogonal contrast to compare individual or groups of means to each other.

10. Results, lines 304-313. If no significant effects were found for Nosema spore abundance for either method or between methods in any year, the initial statement: “Wintering method had effects on Nosema spore abundance” is not accurate and should be reworded.

 Author Response: Agreed. The opening line is now as follows (line 310-311):

 “Nosema spore abundance did differ between wintering methods, but not consistently.”

11. Discussion. Good discussion, but the main issue is to try to infer that wintering method interacted with N. ceranae infections, which resulted in differential survival or population growth of colonies. Your experimental design and statistical methods do not allow reaching that conclusion. It could be said that this is a possible explanation of the results (as a hypothesis rather than as a conclusion), but that further research is needed to separate the effects of wintering method and Nosema spp. infections or demonstrate their interaction on overwinter survival and posterior colony growth. The difference in colony survivorship and development found in this study could be mainly due to the wintering method since you could not demonstrate effects of wintering method on Nosema spp. abundance. You also do not show a correlation between Nosema abundance in winter and mortality or colony development. And no interaction effects between Nosema spp. abundance and wintering method on colony survivorship and growth are shown. Experiments with proper controls (Nosema-free colonies and Nosema-positive colonies treated with fumagillin) would be required in future experiments. Because of the above, I suggest rewording the parts of the discussion where it is inferred that there is a link between wintering method and Nosema spp. infections on colony survivorship and growth. As suggested before, reword it as a potential explanation, not as a conclusion.

 Author Response: The logistic regression statistics were reviewed and we confirmed that the interaction between Nosema abundance and wintering method was not significant (χ2=1.1739, df=1, P=0.279). Therefore, the difference between wintering methods was consistent across all spore loads in this study. We agree that this may not be a link between wintering method and Nosema, as correctly pointed out by the reviewer. We have reworded references to this in the abstract, results, and discussion to indicate it as a possible explanation, rather than a conclusion, and further controlled studies that manipulate Nosema levels are needed to test this. 

 In the abstract, and the following changes were made: 

 Line 12-15- “However, wintering method affected survival with colonies wintered indoors having lower mortality and more rapid spring population build-up than outdoor-wintered colonies.”

 Line 18-20 were deleted- “Furthermore, the results suggest that mitigation of temperature stress associated with indoor wintering decreased the impact of Nosema infections on colonies more than any potential benefits associated with weather allowing late winter cleansing flights.”

 In the results, the following changes were made: 

 Line 358-359- “There was a significant effect of Nosema spore level on overall colony mortality over the almost two-year study.”

 Line 361-365- “Additionally, outdoor-wintered colonies were more likely to die than indoor-wintered colonies (χ2=4.9896, df=1, P=0.026; Fig 6). There was no significant interaction between Nosema abundance and wintering method, indicating that indoor-wintered colonies had increased survival relative to outdoor-wintered colonies across the range of spore levels found in this study.”

 Line 365-367 was removed- “At the one million spores/bee threshold, outdoor-wintered colonies were almost five times as likely to die than indoor-wintered colonies (48.9% and 9.9%, respectively).”

 In the discussion, the following changes were made: 

 Line 412-413- “However, colonies that were wintered indoors had lower mortality and faster spring population build-up than outdoor-wintered colonies.”

 Line 474-484- “The data suggests that indoor-wintered colonies were less likely to die from Nosema infections and have larger bee populations in the following summer than outdoor-wintered colonies. However, our data did not conclusively show this and must be clarified through further studies that manipulate Nosema levels in indoor and outdoor winter environments are needed. Our study supports other Canadian studies that have demonstrated that colonies infected with high levels of parasites and/or pathogens were more likely to survive when wintered indoors than when wintered outdoors [13,25,48]. The apparent difference in colony mortality between wintering methods across all levels of Nosema abundance suggests that the threshold for damage tolerated by beekeepers from Nosema infection should be lower for colonies that will be wintered outdoors to have survival similar to indoor-wintered colonies, which also needs to be confirmed with further studies.”

 Line 488-490- “These results showed that indoor wintering is an attractive option for beekeepers looking to reduce mortality due to Nosema and increase summer colony population size.”

 Line 495-496 were removed- “The results of this study show that the wintering method (indoor versus outdoor) affected the impact of Nosema on colony performance.”

 Line 496-500- “This study found that colonies performed better when wintered indoors than wintered outdoors. These results suggest that mitigation of temperature stress during indoor wintering may have affected the impact of Nosema infections on colonies more than any benefits associated with the availability of late winter cleansing flights.”

 Line 525-532- “There was no consistent effect of wintering method on Nosema abundance; however, indoor wintering was more effective in increasing colony survival than outdoor wintering under the range of Nosema abundances found in this study. Also, indoor-wintered colonies had greater populations in the following spring than their outdoor-wintered counterparts. Therefore, to achieve similar mortality to indoor-wintered colonies, the data suggests that the one million spores/bee threshold as assessed in spring needs to be lowered for colonies being wintered outdoors. However, this requires further research.”

12. Figures 5-8 could be Fig 5 a,b,c,d. Figure 8: line patterns are not clearly different.

 Author Response: We agree that some of the figures could be combined to reduce repetition. Therefore, we combined Fig 5 and Fig 6 into Fig 5 A, B. See line 375-381 for new figure description. We decided to keep Fig 7 and 8 separated as they had less similarities in their figure descriptions. The figures are now Fig 5A and 5B, Fig 6, and Fig 7. References to the figure numbers have been updated to match the changes made.

 We assume that you meant Figure 7, not 8. The lines for Fig 6 (formerly Fig 7) have been changed to a solid color: red for Indoor and blue for Outdoor.

Responses to Reviewer #2: 

In this article the authors were whether Nosema abundance differed between northern and southern locations in Alberta, whether nosema abundance differed in outdoor vs indoor wintered colonies. They followed up with examining bee population buildup, followed colony survival, and economic viability over the almost 2 year period. The finding that N. apis had 4X higher copies than N. ceranae is interesting. In other studies with mixed infections, N. apis is usually at much lower levels – maybe an effect of latitude?

 Author Response: A mixed infection only occurred in the North apiary after it was wintered indoors. Because of the lack of replication, we do not feel confident speculating as to the cause of this.

General comments:

The paper was well written and easy to follow with nice results. The authors are familiar with the literature and presented their findings in context with other studies well.

 Where does the 1 million spore/bee threshold come from? Is this only for N. apis? I am not aware that one had been established for N. ceranae but rather used the same assumption of the N. apis threshold. This may not be appropriate given the differences in impacts on bees by the two species.

 Author Response: This was addressed at line 57-61 and which states:

 “Alberta's current control recommendations are to apply fumagillin in the spring and fall when spore abundance is above one million spores per bee [32]. However, this nominal threshold used throughout North America was established for N. apis infections and has not been appropriately validated for either N. apis or N. ceranae under different beekeeping winter management and climatic conditions.”

Overall the figures are well presented and easy to follow. Figures 7 and 8 could be clarified more. The square and circles at the top and bottom do not seem to mean much as presented. What do the dashed lines represent? It is difficult to determine which is indoor vs outdoor.

 Author Response: The lines for Fig 6 (formerly Fig 7) have been changed to red for Indoor and blue for Outdoor. The squares and circles on the graphs indicate whether colonies were alive (0) or dead (1) at specific spore levels within different wintering methods. It is a standard output of a logistic regression analysis. The following was added for clarification (line 391-392): “The symbols indicate whether colonies were alive (0) or dead (1) at a specific Nosema abundance within wintering method.”

Treatment would be best before the levels peak in the spring, however there are issues with residuals and harvesting the honey.

 Author Response: In Alberta, the honey flow is much later in the spring, so honey supers would not be on at the time of treatment. Therefore, there is minimal risk of residues in extracted honey.

The economic viability aspect was not touched on as much. With “economic viability” I think that I am going to see numeric comparisons. How expensive is it to overwinter colonies? How does that cost compare to the difference in mortality from indoor vs outdoor overwintered colonies? Maybe a recommendation would be to not winter outdoors after the first winter but after the second where there is a significant difference in the number of adult bees.

 Author Response: The term ‘economic viability” is misleading and has been changed to ‘commercial viability’ at line 8 and 99. See further response under Reviewer 1, comment 3.

Specific comments:

Line 81: I have limited knowledge on overwintering bees indoors. Are bees allowed to take “scheduled” defecation flights while indoor?

 Author Response: No, indoor wintered colonies do not take scheduled flights. See line 81-84 which states:

 “In contrast, colonies that overwinter indoors are moved in the autumn (late October) into buildings that are temperature regulated at 4-5°C with constant air exchange and air remixing where the colonies are always kept in the dark to prevent bee flight [36].”

Lines 140-141: Was there a reason for using queens from Kona Queen? Why not use locally adapted queens?

 Author Response: Queens from a single source were used to standardized genetics among the experiment colonies to reduce variability. At the time, there were not many local queen breeders in Alberta, and they often do not have queens available until later in the spring. Kona Queens were readily available in large quantities at the beginning of the experiment. Also, Kona Queens are commonly used by other beekeepers in Alberta.

Lines 419-420: why is treating with fumagillin in October too late in Alberta? Is it because the temperatures are already freezing? Would it be feasible if the colonies were overwintered indoors where there could be a controlled increase in temperature and light to allow them to break cluster to feed?

 Authors Response: In October, freezing temperatures do occur in Alberta. Additionally, feed is generally given in September, so it is convenient for beekeeper to provide treatment in syrup then. Also, bees do not take syrup as readily when temperatures get colder. 

 There are beekeepers that do feed indoor wintered colonies later in the winter if they think they require it. So, it is possible that treatment could be given at this point. However, it may be too late by that point.

 The following explanation has been added to the discussion (line 439-441):

 “By this point, colonies are being winterized due to the rise of freezing temperatures. Additionally, bees do not take syrup as readily at these temperatures, making treatment impossible.”

Additional edits made to the Manuscript:

Fig 1, Fig 2: changed to solid color. North is black and South is yellow.

Fig 3, Fig 4: changed to solid color. Indoor is red and Outdoor is blue.

---

## [Editor Report · Decision Letter 1]

6 Oct 2021

Epidemiology of *Nosema* spp. and the effect of indoor and outdoor wintering on honey bee colony population and survival in the Canadian Prairies

PONE-D-21-14529R1

Dear Dr. Punko,

We’re pleased to inform you that your manuscript has been judged scientifically suitable for publication and will be formally accepted for publication once it meets all outstanding technical requirements.

Kind regards,

Wolfgang Blenau

Academic Editor

PLOS ONE
---

## [Editor Report · Acceptance letter]

15 Oct 2021

PONE-D-21-14529R1 

Epidemiology of *Nosema* spp. and the effect of indoor and outdoor wintering on honey bee colony population and survival in the Canadian Prairies 

Dear Dr. Punko:

I'm pleased to inform you that your manuscript has been deemed suitable for publication in PLOS ONE. Congratulations! Your manuscript is now with our production department. 

Kind regards, 

on behalf of

Dr. Wolfgang Blenau 

Academic Editor

PLOS ONE